# CAPNet: Cartoon Animal Parsing with Spatial Learning and Structural Modeling

Jian-Jun Qiao
Southwest Jiaotong University
Chengdu, China
qjjai56@gmail.com

Meng-Yu Duan
Southwest Jiaotong University
Chengdu, China
duanmengyu369@gmail.com

Xiao Wu*
Southwest Jiaotong University
Chengdu, China
wuxiaohk@gmail.com

Wei Li
Southwest Jiaotong University
Chengdu, China
liwei@swjtu.edu.cn

## Abstract

Cartoon animal parsing aims to segment the body parts such as heads, arms, legs and tails of cartoon animals. Different from previous parsing tasks, cartoon animal parsing faces new challenges, including irregular body structures, abstract drawing styles and diverse animal categories. Existing methods have difficulties when addressing these challenges caused by the spatial and structural properties of cartoon animals. To address these challenges, a novel spatial learning and structural modeling network, named CAPNet, is proposed for cartoon animal parsing. It aims to address the critical problems of spatial perception, structure modeling and spatial-structural consistency learning. A spatial-aware learning module integrates deformable convolutions to learn spatial features of diverse cartoon animals. The multi-task edge and center point prediction mechanism is incorporated to capture the intricate spatial patterns. A structural modeling method is proposed to model the complex structural representations of cartoon animals, which integrates a graph neural network with a shape-aware relation learning module. To mitigate the significant differences among animals, a spatial and structural consistency learning strategy is proposed to capture and learn feature correlations across different animal species. Extensive experiments conducted on benchmark datasets demonstrate the effectiveness of the proposed approach, which outperforms the state-of-the-art methods.

## CCS Concepts

• **Computing methodologies → Image segmentation**.

## Keywords

Cartoon Animal Parsing, Deformable Convolution, Graph Neural Network

*Corresponding author: Xiao Wu

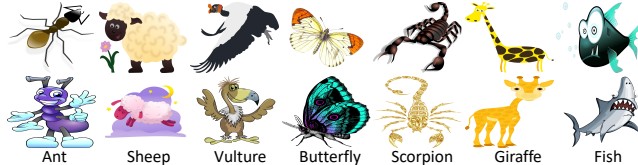

**Figure 1: Cartoon animal parsing is challenging due to irregular body structures, complex visual appearances, abstract drawing styles and substantial variations among different categories. The significant differences of cartoon animals across diverse species and individuals are caused by the abstract spatial and structural properties of cartoon images.**

**ACM Reference Format:**
Jian-Jun Qiao, Meng-Yu Duan, Xiao Wu, and Wei Li. 2024. CAPNet: Cartoon Animal Parsing with Spatial Learning and Structural Modeling. In *Proceedings of the 32nd ACM International Conference on Multimedia (MM '24), October 28–November 1, 2024, Melbourne, VIC, Australia.* ACM, New York, NY, USA, 9 pages. https://doi.org/10.1145/3664647.3680570

## 1 Introduction

Cartoon characters are important components of various multimedia applications with their vivid, attractive and imaginative expressive capabilities. The cartoon-centric applications include the metaverse, animated films, virtual reality and artistic creations. A major category of cartoons is animals, which consist of a wide range of diverse species such as reptiles, birds, mammals and fish. They typically exhibit diverse, rich and complex body parts, such as ears, limbs, wings and claws.

Cartoon animal parsing, an emerging frontier of cartoon animals, seeks to semantically delineate various body parts. It enhances the comprehension of cartoon characters and makes progress in cartoon-centric applications. Unfortunately, previous cartoon parsing methods are proposed to parse anthropomorphic cartoon characters [26] or single animal category like cartoon dog [33], which exhibit more visual and structural consistency. On the contrary, cartoon animals have significant diversity in visual appearances and structures. As a result, techniques of cartoon parsing have unsatisfactory performance in the domain of cartoon animal parsing, which motivates this work.

Cartoon animal parsing is a challenging task due to the irregular structures, complex appearances, abstract styles and various animal

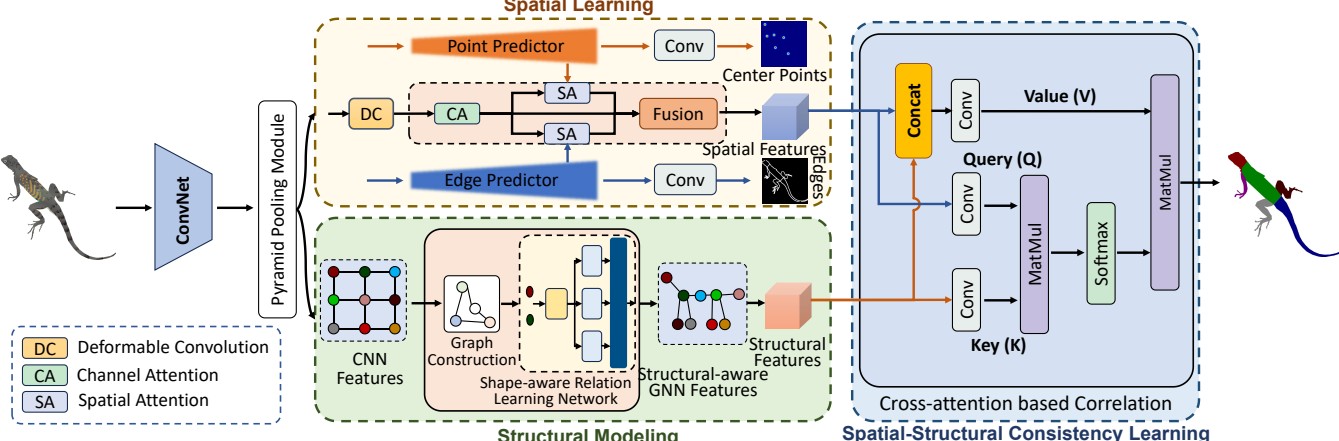

**Figure 2: The framework of the proposed cartoon animal parsing network (CAPNet). It consists of spatial learning, structural modeling and spatial-structural consistency learning. The spatial learning module captures the complex spatial patterns and the structural modeling branch models the intricate structural features of cartoon animals. The spatial-structural consistency learning module addresses the challenges posed by the significant diversity and complexity of cartoon animals by conducting consistency learning.**

categories, which is illustrated in Fig. 1. Compared to real-world humans or animals, the body parts in cartoon animals usually demonstrate inconsistent spatial and structural attributes. This includes complex and uncertain spatial distributions of the center points and edges of the body parts, and irregular structural shapes and sizes of the semantic parts. The spatial and structural differences among diverse cartoon animals are conspicuous even within the animals of the same category. More specifically, the body parts of real-world humans or animals possess consistent spatial and structural attributes. The spatial patterns such as center point positions and edge contexts of their body parts, or the structure properties like the shapes and sizes of their semantic parts, are usually consistent. However, when facing cartoon animals, the spatial distributions of the center points and edges of the body parts, or the structural shapes and sizes of the semantic parts, are not consistent due to the abstract properties of cartoon images. Therefore, cartoon animal parsing remains a challenging task.

Conventional human parsing methods have good performance in extracting visual features and predicting pixel-wise labels for human bodies. Although some approaches have incorporated attention mechanisms and graphical models to capture important spatial cues and structural representations, they are designed for the real-world human bodies, which have limited performance when facing the abstract cartoon images. Cartoon parsing methods are proposed to address the problems caused by the scale variations [33] and irregular body structures [26] of cartoon images. Unfortunately, the spatial learning and structural modeling of cartoon animal parsing have not been fully explored.

In this paper, a novel spatial learning and structural modeling network, named CAPNet, is proposed to alleviate the challenging problems of cartoon animal parsing. The framework is illustrated in Fig. 2. The cartoon animal image is fed into ConvNet [12] followed by Pyramid Pooling Module (PPM) [46] to extract contextual features. The spatial learning structure is designed to capture the spatial patterns of the body parts, with deformable convolutions to learn spatial features of irregular body parts. In addition, it predicts the edges and center points of the body parts to obtain edge-aware

and center-aware spatial features. The structural modeling module is designed to capture the intricate structures of cartoon animals. A graph neural network (GNN) is employed to model the complex structures of cartoon animals. A shape-aware relation learning network is designed to learn the relations of the node features of GNN, which considers the shape information during the structural modeling. To integrate the spatial and structural features and achieve consistency among diverse cartoon animals, a spatial-structural consistency learning strategy based on cross-attention mechanism is proposed. With the novel structures, CAPNet alleviates the challenges of spatial learning, structure modeling, and spatial-structural consistency learning in cartoon animal parsing. Experiments conducted on the cartoon animal parsing and cartoon parsing datasets demonstrate the effectiveness of CAPNet, which outperforms the state-of-the-art approaches. The contributions are listed as follows:

- A novel spatial learning and structural modeling network named CAPNet is proposed for cartoon animal parsing, which addresses the challenges of spatial learning, structure modeling, and spatial-structural consistency learning.

- To capture and learn the complex and inconsistent spatial patterns in diverse cartoon animals, a spatial learning branch with deformable convolutions and a multi-task prediction strategy is designed, which enhances the spatial awareness of the network.

- To model the structural information and capture the intricate structural relationships within cartoon animals, a structural modeling branch is proposed with a graph neural network and a shape-aware relation network.

- A spatial-structural consistency learning strategy is proposed with a cross-attention mechanism, which aims to achieve consistent learning among spatial and structural features and mitigates the problems caused by the complexity and diversity of cartoon animals.

- The proposed method achieves state-of-the-art performance on the cartoon parsing datasets, which demonstrates the effectiveness of the proposed spatial learning and structural modeling network.

## 2 Related Work

### 2.1 Human Parsing

The human parsing methods [4, 18, 31] have made significant progress in recent years, with numerous approaches proposed to parse body parts of humans.

**Spatial Learning.** Spatial information plays a crucial role in parsing tasks by providing crucial information about the relative positions. It facilitates the recognition of complex and ambiguous regions by considering the spatial distribution of the human body, achieving more accurate parsing results. Previous methods [22, 42] enhance the learning ability of spatial patterns by embedding a pose estimation branch into the human parsing framework. MMAN [21] improves the capability of capturing consistent spatial information through local and global learning. CDGNet [20] learns the spatial characteristics of the human body in horizontal and vertical directions to improve the performance of human parsing. Overall, the integration of spatial information enhances the performance of human parsing algorithms.

**Structural Modeling.** In addition to spatial features, structural information is also important for human parsing. To transcend the limitations of conventional CNN-based approaches, previous works have proposed structural modeling modules. These existing approaches have integrated the prowess of convolutional neural networks (CNNs) [12, 30] and graphical models [13, 19, 23, 28] to extract visual features and model structural information. Previous methods [35, 36] introduce graph neural networks (GNNs) to model the regular structure of human body, leveraging the powerful structural reasoning ability of GNNs to achieve promising results in human parsing. Some methods [17, 44] decompose and model the human body with attention mechanisms [5, 15, 16, 37, 38, 40] to learn and capture important structural information, improving the performance of human parsing. However, these techniques are designed for human parsing and they have limited performance when facing the cartoon animal images with irregular body structures, complex visual appearances, abstract drawing styles, and so on.

### 2.2 Cartoon Parsing

In recent years, there has been a growing interest in the field of cartoon parsing. The traditional cartoon parsing approach [39] segments cartoon images into different regions using conventional linear iterative clustering superpixels and adaptive region propagation merging techniques. However, it lacks semantic discrimination for each segmented region. Recently, pioneering works [26, 33] have adapted deep learning-based human parsing techniques to the realm of anthropomorphic cartoon characters and cartoon dogs, achieving commendable results. DFPNet [33] designs a dense multiscale pyramid network to capture multi-scale information of cartoon dogs, addressing the scale issue in cartoon dog parsing. CPNet [26] proposes a pixel and part correlation method to address the irregularities of cartoon characters. By learning pixel-level and part-level correlations, it identifies complex cartoon structures and improve the performance of cartoon parsing. However, these methods have limitations since they primarily focus on human-like forms or single-category cartoon characters such as dogs, ill-equipped to appearance diversity and structural complexity in different cartoon animals like alligators and butterflies.

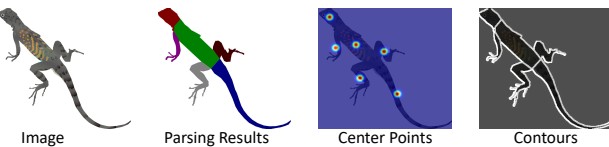

Image     Parsing Results     Center Points     Contours

**Figure 3: Center points and contours of cartoon image.**

## 3 Cartoon Animal Parsing

### 3.1 Framework

To alleviate the challenges of cartoon animal parsing, a novel spatial learning and structural modeling network, named CAPNet, is proposed to integrate spatial perception, structure modeling, and spatial-structural consistency learning. The framework is illustrated in Fig. 2. ConvNet [12] and Pyramid Pooling Module (PPM) [46] are used to extract convolutional features. The spatial learning branch introduces deformable convolutions to learn spatial information related to the irregular body parts of cartoon images. It is augmented by multi-task edge and center point prediction strategy, which captures the edge-aware and center-aware spatial features. The structural modeling branch employs a graph neural network to model the complex cartoon animal structures. It designs a body part relation learning network based on shape-aware convolution and self-attention mechanism, which captures structural shape information and associates the node features of the body parts. The captured spatial features and structural representations are seamlessly integrated with cross-attention mechanism, which achieves consistency among the complex and diverse cartoon animals.

### 3.2 Spatial Learning

The spatial information is of great importance for cartoon animal parsing. The distributions of the center points and edges of the body parts are critical to delineate the spatial patterns of cartoon animal images, which are illustrated in Fig. 3. To learn the intricate spatial patterns, CAPNet integrates deformable convolutions to transcend the rigid receptor fields of conventional convolution, capturing spatial features of abstract cartoon images. It incorporates multi-task edge and center point prediction strategy to capture edge-aware and center-aware spatial features, which makes the network adapt to the distortions and abstraction in spatial dimension. The framework of the proposed spatial learning is illustrated in Fig.4, in which the middle branch with deformable convolutions is adopted to extract spatial features of irregular body parts. The edge prediction module aims to capture discriminative representations of contours, while the center point estimation part provides spatial cues that facilitate the localization and delineation of body parts.

Formally, given features $X \in \mathbb{R}^{C \times H \times W}$ generated by PPM module [46], $X_d$ is the output of the deformable convolutions with $X$ as input. $C$ refers to the number of channels, $H$ and $W$ denote height and width of the feature maps. The edge features $X_e$ are generated by an edge predictor consisting of 5 convolutional layers, with shallow and deep features of the backbone used as inputs [18, 27]. To generate center-aware spatial features $X_h$, $X$ is fed into a heat map decoder consisting of 2 deconvolution layers. $X_d$ is then fed into a channel attention module [15], aiming to build the connections of feature maps in $X_d$. The channel attention-based refinement bridges the gap among the edge prediction task, the center point

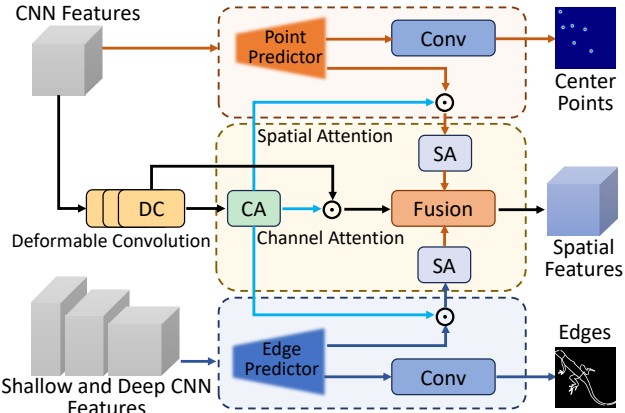

Figure 4: The framework of spatial learning.

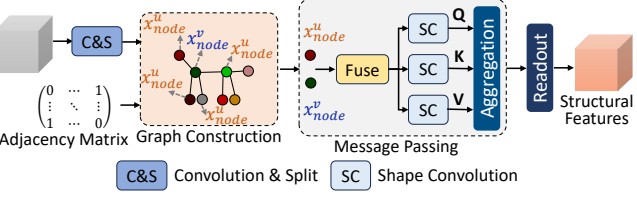

Figure 5: The framework of structural modeling.

prediction task, and the parsing task. It is formulated as,

$$W_c = Sig(Conv(Mp(X_d)))  \quad (1)$$

$$X_d^c = X_d \oplus W_c \odot X_d, X_e^c = X_e \oplus W_c \odot X_e, X_h^c = X_h \oplus W_c \odot X_h \quad (2)$$

where $W_c$ denotes channel attention weights. $Sig(\cdot)$ denotes sigmoid operation. $Conv(\cdot)$ means convolution operation. $Mp(\cdot)$ refers to global max pooling. $\oplus$ and $\odot$ denote element-wise addition and multiplication, respectively.

The proposed approach leverages a spatial attention mechanism [38] to effectively integrate the complementary spatial features from the edge prediction and center point prediction branches. The enriched spatial features are transferred to the parsing branch. This enables the parsing branch to obtain discriminative spatial cues captured by the auxiliary tasks. The synergistic integration of multi-task spatial features, mediated by attention-based fusion, helps the network learn important spatial cues in cartoon images. It is formulated as follows:

$$W_{se} = Sig(Conv(Cat[Max(X_e^c), Avg(X_e^c)])) \quad (3)$$

$$W_{sh} = Sig(Conv(Cat[Max(X_h^c), Avg(X_h^c)])) \quad (4)$$

$$X_e^{sp} = X_d^c \oplus X_e^c \odot W_{se}, \quad X_h^{sp} = X_d^c \oplus X_d^c \odot W_{sh} \quad (5)$$

where $W_{se}$ and $W_{sh}$ indicate the spatial attention weights from edge-aware features and center-aware features, respectively. $Cat(\cdot)$ denotes concatenation operation. $Max(\cdot)$ and $Avg(\cdot)$ are the functions aggregating salient spatial features by computing max and average values alongside the channel dimension, respectively.

Finally, the edge features $X_e^c$, the center point features $X_h^c$, the edge-aware parsing features $X_e^{sp}$ and the canter-aware parsing features $X_h^{sp}$ are integrated to obtain the spatial-aware parsing features. It is formulated as,

$$X_{sp} = Cat[X_e^c, X_h^c, X_e^{sp}, X_h^{sp}] \quad (6)$$

Through the integration of deformable convolutions and multi-task predictions, the network learns spatial awareness. This is indispensable to address the parsing problem caused by the complex spatial distributions of cartoon animals.

## 3.3 Structural Modeling

The structural information, including the structural shapes and relations of the body parts, is also important for cartoon animal

parsing. To model the complex structural features that characterize cartoon animals, the structural modeling module is designed. CAPNet designs a shape-aware graph neural network to model graphical representations of cartoon animals. The graph neural network is adopted to construct the structures of cartoon animals and capture the relations among the body parts. The shape convolutions are utilized to capture shape information for the graph nodes. Moreover, a self-attention mechanism is adopted to associate the graph nodes with shape information.

The framework of the structure modeling is illustrated in Fig. 5. CAPNet leverages the shape-aware GNN to construct and refine a graphical representation $\mathcal{G} = (\mathcal{V}, \mathcal{E})$, where $\mathcal{V}$ denotes the set of nodes corresponding to body parts, and $\mathcal{E}$ encodes the structural relationships of these nodes. The shape-aware GNN operates by updating the node representations of the body parts through a message passing scheme.

In the structure modeling, $X$ is processed to node features with a convolutional layer to adjust channel dimension and a split operation to split the features into node features $X_{node} \in \mathbb{R}^{|\mathcal{V}| \times c \times H \times W}$. $|\mathcal{V}|$ denotes the number of nodes and $|\mathcal{V}| \times c = C$. The adjacency matrix is calculated to establish connections between neighboring semantic parts. CAPNet leverages Message Passing Neural Network (MPNN) [9] to model the correlations among nodes. MPNN is structured into two stages: message passing and readout. CAPNet designs a shape-ware relation network as the message-passing function for cartoon animal features, aiming to connect different nodes with the guidance of shape information. For every node $v$, the message passing gathers messages $m_v$ from its neighbors $\mathcal{N}_v$, which is formulated as,

$$m_v = \sum_{u \in \mathcal{N}_v} M(X_{node}^u, X_{node}^v) \quad (7)$$

where $X_{node}^v$ and $X_{node}^u$ are elements of $X_{node}$. $M(\cdot)$ is the message function used to collect part relations. $M(\cdot)$ is formulated as follows:

$$M(X_{node}^u, X_{node}^v) = \phi(Cat[X_{node}^u, X_{node}^v]) \quad (8)$$

where $\phi$ denotes the shape-ware relation learning network.

The proposed shape-aware relation learning network leverages the information association capability of self-attention, which captures the long-range dependencies of the body parts. It distinguishes itself by incorporating shape-aware convolutions to perceive geometric shapes and utilizing prior geometric information. By combining self-attention with shape-aware convolution and geometric priors, the method aims to capture both long-range dependencies and salient structural representations effectively.

The framework of the shape-aware relation network is illustrated in Fig. 5, which takes the graph nodes as inputs. The graph nodes are fed into three shape-aware convolutions with kernel size of 3, which aims to capture shape information. The vanilla convolution

layer is defined as,

$$X_{vanilla} = Conv(W_K, X_{cat}^{u,v}) \tag{9}$$

where $X_{cat}^{u,v} = Cat[X_{node}^u, X_{node}^v]$. $W_K$ denotes the learnable weights of kernels in a convolution layer.

CAPNet employs the shape-aware convolution [1] to replace the vanilla convolution in the shape-aware relation network, which aims to capture shape information. It is formulated as,

$$X_{sc}^Q = SC(W_K, W_B, W_S, X_{cat}^{u,v}) = Conv(W_K, W_B \diamond X_B + W_S * X_S) \tag{10}$$

where $SC(\cdot)$ denotes shape convolution. $\diamond$ and $*$ are base-product and shape-product operator [1], respectively. $X_{cat}^{u,v}$ is decomposed into two components: $X_B$ and $X_S$. $X_B$ is the mean of $X_{cat}^{u,v}$ and $X_S = X_{cat}^{u,v} - X_B$ is the relative features of $X_{cat}^{u,v}$. $W_B$ and $W_S$ are two learnable weights to separately consume $X_B$ and $X_S$. Similarly, $X_{sc}^K$ and $X_{sc}^V$ are computed as the keys and values for the self-attention module, respectively.

The shape-aware structure learning process is defined as,

$$M(X_{node}^u, X_{node}^v) = SelfAtten(X_{sc}^Q, X_{sc}^K, X_{sc}^V) = Softmax(A + G)X_{sc}^V \tag{11}$$

where $G$ is initialized and learned as [14]. The difference is that the geometric priors of CAPNet are applied for the global feature maps. $A$ is the attention maps calculated from $X_{sc}^Q$ and $X_{sc}^K$ [8, 32].

With the gathered message $m_v$, the node feature $X_{node}^v$ is updated as,

$$X_{up}^v = ConvGRU(X_{node}^v, m_v) \tag{12}$$

where $ConvGRU$ [29] is adopted as update function. In the read-out process, the node features are combined and projected into segmentation features as,

$$X_{st} = Conv(Cat[X_{up}^0, X_{up}^1, ..., X_{up}^{|\mathcal{V}|-1}]) \tag{13}$$

By integrating the GNN-based structure modeling and shape-aware relation learning, the network captures the intricate structural information of diverse cartoon animals. This is of great importance to delineate the complex and diverse body parts.

## 3.4 Spatial-Structural Consistency Learning

The body parts of cartoon animals exhibit substantial variations across different animal categories, which are usually caused by the complex and inconsistent spatial and structural characteristics. Therefore, a novel spatial-structural consistency learning (SSCL) mechanism is proposed to seamlessly integrate the spatial and structural features, achieving consistency among the spatial and structural dimensions. SSCL employs the cross-attention mechanism to correlate the spatial and structural features. It learns a consistent feature representation by fully utilizing the complementary spatial and structural information. Specifically, the spatial information is employed as the queries, while the structural information serves as the keys. The rationale behind this design is to exploit the query-key matching mechanism, which aims to learn the correspondences between spatial and structural features. For diverse cartoon animals, their spatial and structural characteristics may vary significantly. The cross-attention mechanism can adaptively adjust the fusion

weights between these two modalities, effectively capturing the correlations between spatial and structural information. This results in a more consistent feature representation that better encapsulates the holistic characteristics of the cartoon animals.

In the consistency learning, the spatial features $X_{sp}$ are treated as the queries, the structural features $X_{st}$ are used as the keys, and the concatenated features $X_{cat} = cat[X_{sp}, X_{st}]$ as the values. The consistency learning is formulated as follows:

$$Q = X_{sp}W_q, \quad K = X_{st}W_k, \quad V = X_{cat}W_v \tag{14}$$

$$X_{fused} = CrossAtten(Q, K, V) \tag{15}$$

where $W_q, W_k, W_v$ are learnable weights from convolution layers. Through this aggregation mechanism, a fused feature representation $X_{fused}$ that incorporates consistent spatial and structural information is obtained.

## 3.5 Loss Functions

The objective function of this paper consists of parsing loss, center point loss and edge loss. The parsing loss is set to predict the segmentation results. The center point loss is used to predict the center points of the body parts. The edge loss is utilized to predict the edges of the body parts. The objective function is defined as,

$$\mathcal{L}_{all} = \mathcal{L}_{parsing} + \mathcal{L}_{spatial} + \mathcal{L}_{edge} \tag{16}$$

where $\mathcal{L}_{edge}$ [20, 27] denotes the weighted cross-entropy loss between the detected edge map and the binary edge label map.

The parsing loss is defined as,

$$\mathcal{L}_{parsing} = -\frac{1}{M}\sum_i^M\sum_n^N g_{in}logp_{in} \tag{17}$$

where $g_{in}$ denotes the ground truth label of the $n$-th body part on the $i$-th pixel and $p_{in}$ is the related prediction result. $M$ is the number of pixels in the image. $N$ is the number of classes.

The center point loss [10, 26] is defined as,

$$\mathcal{L}_{spatial} = \frac{1}{N}\sum_n^N\sum_{x,y}\|H_n(x,y) - G_n(x,y)\|_2 \tag{18}$$

where $H_n(x,y)$ represents the predicted result for the $n$-th center point of the $n$-th body part at the pixel location $(x,y)$. $G_n(x,y)$ is the ground truth.

The proposed method alleviates the problems caused by the complexity and diversity in the cartoon animal domain with the above objective function.

## 4 Experiments

### 4.1 Datasets and Evaluation Metrics

**CASet.** To evaluate the proposed methods, a cartoon animal parsing dataset called CASet is collected from the Internet and annotated. It contains 2,643 images of cartoon animals. The dataset is divided into 1,718 images for training, 262 images for validation and 663 images for testing. The images are annotated according to different body parts, which include 9 categories: head, body, wing, tail, left arm, right arm, left leg, right leg, and a background class. The dataset comprises a wide range of cartoon animal categories, including 52 classes such as ant, horse, monkey and penguin. Each category contains dozens of cartoon images. The 52 animal categories are listed as follows: alligator, antelope, ant, bee, butterfly,

Jian-Jun Qiao, Meng-Yu Duan, Xiao Wu, and Wei Li

**Table 1: Comparison on CASet.**

| Method | Pixel Acc. | Mean Acc. | Mean IoU |
|---|---|---|---|
| DeepLabV3+ [3] | 93.71 | 80.05 | 69.74 |
| DFPNet [33] | 94.38 | 80.81 | 70.74 |
| CE2P [27] | 94.30 | 81.07 | 70.81 |
| HHP [36] | 94.08 | 80.23 | 70.84 |
| SCHP [18] | 94.37 | 82.24 | 72.32 |
| CDGNet [20] | 94.56 | 83.02 | 73.43 |
| CPNet [26] | 94.67 | 83.28 | 73.58 |
| CAPNet (Ours) | **94.86** | **84.33** | **74.57** |

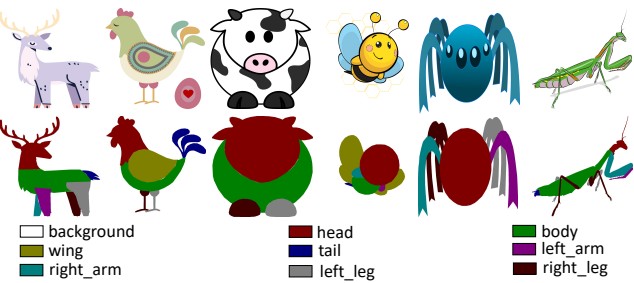

background  head  body
wing  tail  left_arm
right_arm  left_leg  right_leg

**Figure 6: Visualizations of CASet.**

cat, chameleon, chicken, chimpanzee, cicada, cockroach, cow, crab, deer, dinosaur, dog, dragonfly, duck, dung beetle, elephant, fly, frog, fox, gecko, giant panda, giraffe, hawk, hippopotamus, horse, human, lion, lizard, mantis, monkey, mosquito, mouse, ostrich, penguin, pigeon, pig, polar bear, rabbit, scorpion, sheep, shrimp, sloth, spider, squirrel, tiger, turtle, vulture and wolf.

**CartoonSet.** The CartoonSet dataset [26] consists of 2,229 cartoon images. Among these images, 1,530 are used for training, 510 for testing, and 189 for validation. This dataset encompasses a wide range of cartoon styles, including drawings of children, illustrations with brief strokes, and animated characters. Each image contains detailed annotations of 24 classes for the body parts.

**Cartoon dog.** The Cartoon dog dataset [33] has 965 images of diverse cartoon dogs. It is divided into 773 images for training and 192 images for testing. The Cartoon dog dataset provides annotations of eight classes for the cartoon images.

**LIP.** The LIP dataset [10] is a human parsing dataset containing 50,462 single-person images. It is divided into 30,462 images for training, 10,000 for testing, and an additional 10,000 for validation. 19 classes of the body parts of humans are annotated.

**Evaluation Metrics.** Following previous studies [20, 25, 26, 33], Mean Intersection over Union (Mean IoU) is adopted as the evaluation metric. It computes the average intersection-over-union ratio between predicted body parts and ground truth. Moreover, Mean Accuracy (Mean Acc.) is utilized for per-class accuracy calculation, and Pixel Accuracy (Pixel Acc.) measures the accuracy of correctly predicted pixels.

### 4.2 Implementation Details

To implement the proposed CAPNet, a ResNet-101 backbone [12] with PPM [46] and DeepLabv3+ decoder [3] is used as the baseline model. Stochastic Gradient Descent (SGD) is employed as the optimizer with a momentum of 0.9 and a weight decay of 5e-4. Following previous works [2, 8, 24, 43], CAPNet employs the "poly" learning rate strategy, defined as $lr = lr_{init} \times (1 - \frac{C_n}{T_n})^{power}$, where $lr$ and $lr_{init}$ are the current learning rate and the base learning rate, respectively. $power = 0.9$. $C_n$ and $T_n$ represent the current iteration number and the total iteration number, respectively. For data augmentation, CAPNet applies random left-right flipping with a 0.5 probability and random scaling strategy.

For a fair comparison on CASet, all methods are trained with a batch size of 8 and image size of 384×384. The number of training epochs is 150. For the compared methods, other train settings are kept the same as their original papers. For CartoonSet, CAPNet is trained with 150 epochs. The learning rate is 7e-3 and the training size of images is 384×384. For Cartoon dog dataset, CAPNet uses a learning rate of 3e-3, and the training images are resized to 384×384. The number of the training epochs is 300. All methods undergo single-scale evaluation on CASet, CartoonSet, and Cartoon dog datasets. For LIP dataset, CAPNet is trained with a learning rate of 7e-3. The training images are resized to 473×473. The number of the training epochs is 150. For a fair comparison, CAPNet utilizes the multi-scale evaluation approach [18, 20, 36] for the LIP dataset.

### 4.3 Comparisons with the State-of-the-Art Approaches

To evaluate the performance of the proposed method, it is compared with the state-of-the-art cartoon parsing and human parsing methods. The comparison results are listed in Table 1. The general semantic segmentation method DeepLabV3+ [3] lacks dedicated modules for cartoon animal parsing. Accordingly, its performance is limited. Human parsing methods such as CDGNet [20], CE2P [27], and SCHP [18] focus on spatial information extraction and label noise learning, which improves the accuracy of cartoon animal parsing. But they ignore the modeling and learning of structural information, which limits the performance. HHP considers structural information modeling but overlooks spatial information learning. Moreover, when capturing spatial and structural information, they mainly consider the simple spatial patterns and regular structures of the human body, without taking into account the abstract and irregular properties of cartoon animals. DFPNet [33] is used for cartoon dog parsing with a dense pyramid learning structure to address the multi-scale problem in cartoon parsing. But its performance is limited when facing the challenges posed by cartoon animal parsing. CPNet [26] improves the segmentation of the irregular structures of cartoon characters using pixel and part correlation learning strategy. However, it focuses on independent cartoon characters and has limited generalization ability when segmenting diverse cartoon animals. Overall, these existing methods ignore the intrinsic differences between cartoon animals and real-world humans. The distinctions between cartoon animals and common cartoon characters are also not noticed. As a result, they have difficulties in cartoon animal parsing. To address the challenges in cartoon animal parsing, CAPNet captures the important spatial information and complex structural context by introducing the spatial learning method and structural modeling approach. It deeply explores the essential characteristics of cartoon characters by associating the spatial and structural features, achieving consistency among diverse and complex cartoon animals. The proposed method achieves the highest results on CASet, outperforming the compared methods.

Table 2 gives the comparison results on the CartoonSet dataset. CartoonSet is a cartoon parsing dataset containing diverse and multi-style cartoon characters. Previous methods have a similar

**Table 2: Comparison on CartoonSet.**

| Method | Pixel Acc. | Mean Acc. | Mean IoU |
|---|---|---|---|
| DeepLabV3+ [3] | 87.28 | 63.42 | 50.12 |
| DFPNet [33] | 88.21 | 65.33 | 51.71 |
| HHP [36] | 87.51 | 64.42 | 51.98 |
| CE2P [27] | 88.06 | 66.55 | 52.90 |
| CNIF [35] | 87.74 | 66.01 | 53.21 |
| CDGNet [20] | 88.11 | 67.98 | 53.99 |
| SCHP [18] | 88.63 | 69.36 | 55.44 |
| CPNet [26] | 89.42 | 69.90 | 57.02 |
| CAPNet (Ours) | **89.51** | **70.83** | **57.61** |

**Table 3: Comparison on Cartoon dog dataset.**

| Method | Pixel Acc. | Mean Acc. | Mean IoU |
|---|---|---|---|
| Mask R-CNN [11] | 89.21 | 57.78 | 50.56 |
| CE2P [27] | 92.63 | 76.43 | 65.32 |
| DFPNet [33] | 93.50 | 79.40 | 68.39 |
| SCHP [18] | 94.05 | 81.15 | 71.22 |
| CDGNet [20] | 94.09 | 80.03 | 71.44 |
| CPNet [26] | 94.32 | 82.60 | 72.28 |
| CAPNet (Ours) | **94.58** | **83.78** | **74.30** |

trend as in CASet. Overall, approaches using conventional deep learning methods have relatively worse performance, for example, DeepLabV3+ [3]. When structural representations are captured with GNN, the performance is boosted, for example, HHP [36], CNIF [35] and CPNet [26]. Methods like CE2P [27] and CDGNet [20] incorporate the spatial information, including edges and class distributions, which further improves the performance of cartoon parsing. But when facing the irregularity and complexity of cartoon characters, the performance of previous methods is limited. CAPNet outperforms the state-of-the-art methods by learning the important spatial cues like center points and contours, and crucial structural shapes and relations of the body parts.

The results of different methods on the Cartoon dog dataset are listed in Table 3. The Cartoon dog dataset contains cartoon dogs of diverse styles and appearances, for which traditional segmentation methods like Mask R-CNN [11] and human parsing approaches like CE2P [27] are not well-suited. DFPNet [33] is proposed specifically for cartoon dog parsing and improves the results, but it primarily focuses on multi-scale feature learning. The performance is boosted by CDGNet [20] with its class distribution learning module. SCHP [18] further improves the results by proposing a self-correction strategy to correct misclassified regions. CPNet [26] learns the irregularity of cartoon characters, achieving better results. However, these methods still overlook the unique spatial and structural characteristics of diverse cartoon dogs, resulting in limited performance. Compared to these methods, the proposed CAPNet leverages the essential characteristics of cartoon dogs by focusing on the diversity and complexity of cartoon images, which achieves the state-of-the-art parsing results on the Cartoon dog dataset.

To evaluate the generalization ability of the proposed method for human parsing, the comparison of different methods on LIP dataset is listed in Table 4. LIP is a large-scale human parsing dataset that is widely used [18, 20]. DeepLabV3+ [3], OCR (ResNet101) [41], HRNetV2 [34] and OCR (HRNetV2-W48) [41] are common semantic segmentation methods. They have limited performance on LIP dataset due to the lack of specific components for human parsing. HHP [36], SCHP [18], CDGNet [20] and similar approaches

**Table 4: Comparison on LIP dataset.**

| Method | Pixel Acc. | Mean Acc. | Mean IoU |
|---|---|---|---|
| DeepLabV3+ [3] | n/a | n/a | 52.09 |
| CE2P [27] | 87.37 | 63.20 | 53.10 |
| CorrPM [45] | 87.68 | 67.21 | 55.33 |
| OCR (ResNet101) [41] | n/a | n/a | 55.60 |
| HRNetV2 [34] | n/a | n/a | 55.90 |
| OCR (HRNetV2-W48) [41] | n/a | n/a | 56.65 |
| CPNet [26] | 88.29 | 68.41 | 57.21 |
| CNIF [35] | 88.03 | 68.80 | 57.74 |
| HHP [36] | **89.05** | 70.58 | 59.25 |
| SCHP [18] | n/a | n/a | 59.36 |
| CDGNet [20] | 88.86 | **71.49** | **60.30** |
| CAPNet (Ours) | 88.40 | 68.80 | 57.57 |

**Table 5: Effect of the proposed modules on CASet dataset. SLB: Spatial Learning Branch. SMB: Structural modeling Branch. SSCL: Spatial-Structural Consistency Learning.**

| Method | Pixel Acc. | Mean Acc. | Mean IoU |
|---|---|---|---|
| Baseline | 93.69 | 79.38 | 69.33 |
| Baseline + SLB | 94.56 | 82.54 | 72.24 |
| Baseline + SMB | 94.46 | 81.84 | 71.97 |
| Baseline + SLB + SMB | 94.82 | 83.52 | 74.04 |
| Baseline + SLB + SMB + SSCL | **94.86** | **84.33** | **74.57** |

are designed with specific modules tailored for human parsing, achieving good performance in human parsing. Although CAPNet is proposed for cartoon animal parsing, it achieves competitive results on LIP dataset, outperforming the cartoon parsing method CPNet [26] and most of the human parsing methods. The general semantic segmentation methods such as HRNetV2 [34] and OCR (HRNetV2-W48) [41], although have good performance on generic datasets like Cityscapes [6], exhibiting limited results on the human parsing task. Compared to these common segmentation methods, CAPNet, which is designed for cartoon animal parsing, achieves better results on LIP dataset. The competitive performance of CAPNet on LIP dataset suggests that the proposed method is effective for human parsing as well and demonstrates its generalization ability from cartoon animal parsing to human parsing.

## 4.4 Ablation Studies

To validate the effectiveness of each module in CAPNet, ablation studies are conducted. Table 5 gives the experimental results of the proposed three modules: SLB (Spatial Learning Branch), SMB (Structural Modeling Branch), and SSCL (Spatial-Structural Consistency Learning). The baseline model, which lacks specialized designs for cartoon animal parsing, has limited performance on CASet dataset. When SLB module is added to learn spatial patterns, a significant performance boost is obtained, which proves the importance of spatial learning. With the SMB adopted, the performance is substantially improved, which demonstrates the crucial role of structural shape and relation modeling. The performance is further enhanced by integrating the spatial features and structural contexts, which indicates the complementary nature of these two types of information to collectively improve cartoon animal parsing. SSCL is proposed to learn the consistency between spatial cues and structural representations, which results in an additional performance gain. This suggests that the integration and consistency learning of the spatial and structural features is essential to address the challenges in cartoon animal parsing.

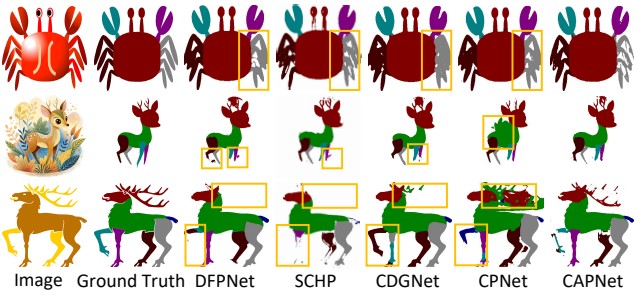

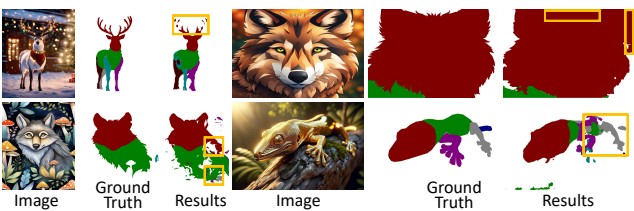

Figure 9: Examples of failure cases.

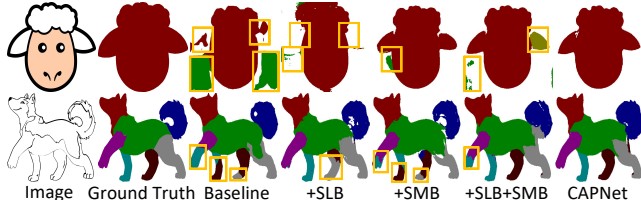

Figure 7: Visualizations of the state-of-the-art methods.

Figure 8: Visualizations of different components.

**Table 6: Effect of the deformable convolutions (DC), shape-aware convolutions (SC), and geometry prior (GP). The symbol $w/o$ means without.**

| Method | Pixel Acc. | Mean Acc. | Mean IoU |
|---|---|---|---|
| CAPNet | **94.86** | **84.33** | **74.57** |
| CAPNet ($w/o$ SC) | 94.78 | 83.48 | 73.85 |
| CAPNet ($w/o$ DC) | 94.57 | 83.18 | 73.22 |
| CAPNet ($w/o$ GP) | 94.75 | 83.53 | 73.72 |

of CAPNet in segmenting diverse and complex cartoon animals is attributed to its capabilities of learning the spatial patterns and modeling the structural representations of cartoon animals.

The visualization results of the proposed modules are illustrated in Fig. 8. Due to the significant differences and complex structures of the cartoon animals, the baseline model struggles to segment the body parts. By incorporating the spatial learning and structural modeling method, the parsing results of cartoon animals are noticeably improved. However, simply combining the spatial learning and structural modeling still leads to limited performance. By leveraging the spatial-structural consistency learning to learn the spatial and structural consistency, the performance of the network is further improved, leading to more accurate parsing results for cartoon animals.

Although CAPNet achieves promising performance for cartoon animal parsing, there is still some space to improve. Currently, it focuses on spatial learning and structural modeling of cartoon animals. But it ignores the interference and influence of complex backgrounds. When the backgrounds have similar colors, visual appearances or textures to cartoon animals, CAPNet has difficulties in capturing and learning the relationship between cartoon animals and the complex backgrounds. As a result, it cannot distinguish cartoon animals from complex backgrounds. Some failure examples are illustrated in Fig. 9, which illustrate the limitations of CAPNet.

To effectively capture the spatial patterns and structural representations of cartoon animals, CAPNet employs deformable convolution [7], shape-aware convolution [1] and geometry prior [14]. Ablation studies are further conducted to demonstrate the effectiveness of these components. The results of the ablation experiments are listed in Table 6. As can be seen from the table, CAPNet without the deformable convolution, shape-aware convolution or geometry prior leads to a decline in performance, which indicates that these components are effective for the network to capture and learn spatial distributions and structural contexts. Deformable convolution, with its unique convolution design, enables the network to learn irregular shapes and structures. It enhances the ability of CAPNet to capture spatial patterns of cartoon images. Shape-aware convolution helps the network capture the shape information of the body parts, which boosts the structural representation learning of CAPNet. The geometry prior strengthens the perception and learning of the geometric structure of cartoon animals. The integration of these components contributes to the state-of-the-art performance of CAPNet in cartoon animal parsing by strengthening the learning and modeling of the important spatial cues and structural contexts.

## 5 Conclusion

In this paper, a spatial learning and structural modeling network, named CAPNet, tailored for cartoon animal parsing is introduced. It addresses the challenges in spatial information learning, structural representation modeling, as well as spatial-structural consistency learning in cartoon animal parsing. The spatial-ware learning structure utilizes deformable convolutions to capture spatial features of irregular body parts. In addition, it predicts edges and center points of semantic parts to learn edge-aware and center-aware spatial patterns. The structural modeling approach incorporates GNN and shape-aware relation learning network to model intricate structures. The spatial and structural consistency learning module enhances feature representation consistency among dissimilar cartoon animals, which demonstrates improved parsing accuracy. In the future, the proposed method will be improved to address the problems caused by complex backgrounds. The dataset will be expanded to include more animal categories with more diverse and intricate backgrounds.

### 4.5 Qualitative Results

Fig. 7 showcases the parsing results of different methods on CASet dataset. Compared to other methods, the proposed method has better performance in segmenting various and intricate body parts of cartoon animals such as crab and deer. The results demonstrate that the proposed method has good generalization ability across different styles and species of cartoon animals. The performance

## Acknowledgments

This work was supported in part by the National Natural Science Foundation of China (Grant No. 62372387, 62001400), Key R&D Program of Guangxi Zhuang Autonomous Region, China (Grant No. AB22080038, AB22080039), Sichuan Science and Technology

Program (Grant No. 2024NSFSC0494), Fundamental Research Funds for the Central Universities (Grant No. 2682024ZTPY044), and China Postdoctoral Science Foundation (Grant No. 2021M702713).

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
