# OpenReview forum: "CAPNet: Cartoon Animal Parsing with Spatial Learning and Structural Modeling"
_acmmm.org/ACMMM/2024/Conference — MM2024 Poster_

### Official Review · Reviewer_zp3J · 2024-04-29

**Rating:** 5
**Confidence:** 3

**Summary:**

This paper focuses on the cartoon animal parsing task, which is difficult since body structures can be irregular and the category can be diverse. To tackle these issues, this paper proposes a new approach with spatial learning and structural modeling. Moreover, the multi-task paradigm is deployed to explore the power of edge and center point detection. Experimental results confirm the effectiveness of the proposed approach.

**Strengths:**

1.	The analysis of the cartoon animal parsing task, including its difficulty and key problems, is well explained in detail, which can provide a good reference for the community.
2.	The designed two branches of architecture, including the spatial learning branch and the structural modeling branch, can serve different roles and provide mutual help for each other, which is also validated by the ablation study.
3.	Experimental results show that the proposed method can achieve the best performance among four challenging datasets.

**Limitations:**

1.	Minor: the template used by this submission is not the latest one. The MM official updated it before the final paper submission deadline.
2.	The limitations of this newly proposed method and future work are not presented.
3.	It is recommended to provide experimental results of different methods’ generalization ability evaluation. Like training on dataset 1 and evaluating on dataset 2 without finetuning.
4.	It is recommended to keep the abbreviation of Table and Figure the same.
5.	Please consider releasing the code for the community.

**Suitability:**

3

---

### Official Review · Reviewer_7m6c · 2024-05-24

**Rating:** 4
**Confidence:** 2

**Summary:**

This paper focuses on the task of cartoon animal parsing.

**Strengths:**

1. The paper is well-written, with the equations, tables, and figures appropriately arranged.
2. The qualitative results look promising; for instance, the segmentation of the wolf head in Figure 7 is very precise and error-free.

**Limitations:**

Cartoon animal parsing is a valuable research direction with significant application potential. I noticed that the experiments in the paper (including the supplementary material) are based on a specific dataset. I wonder if the authors could provide segmentation results on well-known cartoon characters, applying their method to more general and challenging environments such as Peppa Pig, Pokemon, or even anime human characters. Demonstrating the method's effectiveness in these broader contexts could showcase its wider applicability and value.

**Suitability:**

3

---

### Official Review · Reviewer_Qqfn · 2024-05-28

**Rating:** 4
**Confidence:** 3

**Summary:**

The paper introduces CAPNet, a network designed for the task of cartoon animal parsing, which involves segmenting cartoon animal images into distinct body parts like heads, legs, and tails. Unlike existing methods, CAPNet tackles unique challenges posed by cartoon imagery, such as irregular body structures and abstract styles. The network combines spatial learning with structural modeling through a multi-pronged approach. It incorporates deformable convolutions to adapt to the varied shapes of cartoon characters and utilizes a graph neural network alongside a shape-aware relation learning module for understanding complex structural relationships. Extensive experiments demonstrate that CAPNet outperforms existing methods on benchmark datasets.

**Strengths:**

- The topic of cartoon segmentation is intriguing.
- The paper provides a thorough evaluation of the network, including comparisons with state-of-the-art methods across various datasets.
- The paper is generally easy to follow.

**Limitations:**

- In Eq. 8, are there any balancing factors in the loss function? If so, how to determine such hyperparameters?
- It seems to me that the training dataset is quite small-scale. Is there any over-fitting issue occurred during training?
- The improvement over other approaches seems not significant, which could be understandable given the nature of cartoon images.
- It would be great if the authors could include some failure cases and more limitation discussions.

**Suitability:**

3

---

### Meta-Review · Area_Chair_bCdP · 2024-06-30

**Recommendation:** Accept (Poster)
**Confidence:** 5

**Metareview:**

This paper focuses on the cartoon animal parsing task, which is challenging due to the irregular body structures and diverse categories. The topic is very interesting, and the paper provides a thorough evaluation of the network. The qualitative results look promising, and the designed two-branch architecture is reasonable. Although there are some typos, they have been well addressed and do not affect the overall readability. The paper ultimately received scores of Borderline Accept, Borderline Accept, and Weak Accept. Therefore, I recommend this paper to be accepted as a poster for the MM conference.